# Monkeying around with MAIT Cells: Studying the Role of MAIT Cells in SIV and Mtb Co-Infection

**DOI:** 10.3390/v13050863

**Published:** 2021-05-08

**Authors:** Ryan V. Moriarty, Amy L. Ellis, Shelby L. O’Connor

**Affiliations:** Department of Pathology and Laboratory Medicine, University of Wisconsin-Madison, Madison, WI 53711, USA; rmoriarty@wisc.edu (R.V.M.); ellis2@wisc.edu (A.L.E.)

**Keywords:** MAIT, Mtb, SIV, NHP

## Abstract

There were an estimated 10 million new cases of tuberculosis (TB) disease in 2019. While over 90% of individuals successfully control *Mycobacterium tuberculosis* (Mtb) infection, which causes TB disease, HIV co-infection often leads to active TB disease. Despite the co-endemic nature of HIV and TB, knowledge of the immune mechanisms contributing to the loss of control of Mtb replication during HIV infection is lacking. Mucosal-associated invariant T (MAIT) cells are innate-like T cells that target and destroy bacterially-infected cells and may contribute to the control of Mtb infection. Studies examining MAIT cells in human Mtb infection are commonly performed using peripheral blood samples. However, because Mtb infection occurs primarily in lung tissue and lung-associated lymph nodes, these studies may not be fully translatable to the tissues. Additionally, studies longitudinally examining MAIT cell dynamics during HIV/Mtb co-infection are rare, and lung and lymph node tissue samples from HIV+ patients are typically unavailable. Nonhuman primates (NHP) provide a model system to characterize MAIT cell activity during Mtb infection, both in Simian Immunodeficiency Virus (SIV)-infected and SIV-naïve animals. Using NHPs allows for a more comprehensive understanding of tissue-based MAIT cell dynamics during infection with both pathogens. NHP SIV and Mtb infection is similar to human HIV and Mtb infection, and MAIT cells are phenotypically similar in humans and NHPs. Here, we discuss current knowledge surrounding MAIT cells in SIV and Mtb infection, how SIV infection impairs MAIT cell function during Mtb co-infection, and knowledge gaps to address.

## 1. Introduction: Why MAIT a Big Deal about It?

While research into how the immune system responds to HIV and Tuberculosis (TB) has historically focused on CD4+ and CD8+ T cells, unconventional T cells are emerging as interesting targets for immunologic studies. These unconventional T cells, including mucosal-associated invariant T (MAIT) cells, are important in the innate immune response to viral and bacterial pathogens [1,2]. MAIT cells can become activated directly through a highly conserved T cell receptor (TCR) [3,4] or indirectly through cytokine-mediated mechanisms [5,6], which means they can respond quickly and broadly to various pathogens.

MAIT cells are particularly intriguing in the context of HIV and *Mycobacterium tuberculosis* (Mtb) infection because they reside in mucosal sites, such as the rectal mucosa and lungs [7,8,9], where the host initially encounters HIV and Mtb, respectively. However, these tissue sites are often difficult to access in human studies, necessitating the use of animal models. In particular, nonhuman primates (NHPs) are attractive models for HIV/Mtb co-infection because they are naturally susceptible to Mtb [10,11] and Simian immunodeficiency virus (SIV), a NHP version of HIV. By utilizing NHPs in studies of MAIT cell function during SIV, Mtb, and SIV/Mtb coinfection, researchers can examine the specific cellular responses in tissues, both cross-sectionally and longitudinally. In this review, we discuss the importance of MAIT cells, how they are detected, and how they function during HIV, Mtb, and HIV/Mtb coinfection. Further, we discuss how NHP models of SIV, Mtb, and SIV/Mtb have been used to improve our understanding of MAIT cell dynamics longitudinally and within inaccessible tissues affected by SIV (i.e., lymph nodes) and Mtb (i.e., lungs and granulomas).

## 2. Cross-Species Conservation of MR1: Implications for the Study of MAIT Cell Function in Animal Models of Human Disease

Mucosal-associated invariant T (MAIT) cells, initially described in 1993 [12], are innate-like T cells that recognize bacterial riboflavin metabolites presented through the MHC class 1b-related molecule receptor, MR1 [7,9,13]. The MR1 gene is highly conserved across placental mammals [4], with MR1 sequences from NHPs being greater than 95% identical to the human MR1 sequence [14]. Consequently, the MR1 receptor and its interaction with the T cell receptor (TCR) of a MAIT cell may have evolutionarily important immunological functions across many species.

The development of the human MR1 tetramer [7,15] has greatly improved the detection of MAIT cells [8,16,17,18]. Conventionally, MHC: peptide tetramers can be used to label and detect antigen-specific T cells by flow cytometry. Since the MR1 molecule presents metabolites, rather than peptides, the tetramer molecules are folded with the highly stimulatory 5-OP-RU metabolite, which is generated during the riboflavin synthesis pathway [19]. Since the tools to quantify MAIT cells became available, scientists have further characterized the role of MAIT cells in several human diseases, particularly during the course of HIV and TB [17,18,20]. MAIT cells often downregulate the C-type lectin receptor CD161, a molecule historically used to identify MAIT cells along with the TCRVɑ7.2 molecule [6,9,21,22,23,24,25], so the availability of the MR1 tetramer has provided researchers with a more consistent method of MAIT cell detection.

NHP-specific MR1 tetramers were generated, because human MR1 tetramers are not universally cross-reactive because there are amino acid differences in the MR1 molecules found in humans and NHPs in regions vital for proper TCR binding [14]. The availability of NHP-specific MR1 tetramers led to improved detection and characterization of MAIT cells in NHP studies [14,26,27]. Further, this increase in reagent availability has led to more consistent detection of NHP MAIT cells in blood and tissues, which has improved the ability to compare studies across research groups.

## 3. Phenotypic Characteristics of MAIT Cells in Humans and NHPs

NHP MAIT cells are phenotypically and functionally similar to human MAIT cells [9,27]. Because there is no single marker used to identify MAIT cells, they are typically defined using a combination of markers. Both human and macaque MAIT cells commonly express CD8ɑ alone as a homodimer, or both CD8ɑ and CD8β on their surface [7,24,28]. They may also be CD4−CD8−(DN) or CD4+CD8−, but not CD4+CD8+, T cells [29,30,31]. A key marker in the characterization of human and macaque MAIT cells is the high expression of CD161 [6,9,23] and, most commonly, a semi-invariant T cell receptor comprised of Vɑ7.2, also referred to as TRAV1-2, and Jɑ33 [7,12,22,32]. While MAIT cells were initially believed to have little TCR diversity, utilization of the MR1 tetramer has led to the identification of additional TCR combinations [7,33].

MAIT cells in the peripheral blood either an effector memory (CD28−CD95+CCR7−), in both humans and NHPs, or central memory (CD28+CD95+CCR7+) phenotype, unique to NHPs [2,14,26], indicating that they are mature and can readily produce proinflammatory cytokines, such as IFNɣ and TNFɑ, upon antigenic stimulation. Circulating MAIT cells also commonly express tissue-homing markers such as CCR6, CCR5, and ɑ4β7 [9,21,22,34,35] as well as tissue-resident markers such as CD69 and CD103 [9]. The expression of these markers indicates their trafficking to the intestine, liver, and lung, and is similar in both humans [31] and NHPs [27,36]. MAIT cells that traffic to tissues become important in the surveillance of riboflavin-synthesizing bacterial pathogens, such as Mtb, in the periphery and at mucosal sites, including the gut and airways [31], and are highly prevalent in the liver [2,21,27].

## 4. Mechanisms of Human and NHP MAIT Cell Activation

MAIT cell activation occurs through both TCR-dependent interactions with MR1 and TCR-independent cytokine stimulation [37,38,39]. Both methods induce MAIT cell cytokine production and the upregulation of markers on MAIT cells that are consistent with immune activation and degranulation, but vary in the speed of response and types of microbes that utilize each pathway. TCR-dependent activation occurs through the presentation of riboflavin metabolites by MR1 to the MAIT cell TCR. This process requires that the intracellular microbes synthesize riboflavin, and often induces a more rapid response than TCR-independent activation [38,40]. TCR-independent activation, however, occurs through IFNɑ, IL-12, and IL-18 stimulation by antigen-presenting cells, is characterized by a slower response, and allows for MAIT cell activation by non-riboflavin metabolite-producing bacteria, fungi, or viruses [6,38,41,42].

Activated MAIT cells express a combination of markers, including CD40L, CD25, CD69, and PD1 on the cell surface, similar to other activated T cells [24,28,30,41]. Expression of these markers is associated with MAIT cell production of cytokines such as TNFɑ, IFNɣ, IFNɑ, IL-17, and IL-22 [26,31,43]; however, we do not currently know precisely which activation markers expressed by the MAIT cells are associated with the production of specific cytokines.

IL-7, a cytokine important for MAIT cell function [9,44], has been investigated as a treatment to enhance MAIT cell activation during infection with viral pathogens. The addition of IL-7 in vivo and in vitro has been shown to prepare MAIT cells to respond to HIV/SIV infection and improve their function by increasing cytokine production and degranulation [44,45,46]. More specifically, MAIT cells stimulated with IL-7 in vitro has led to upregulation of the degranulation marker CD107a and secretion of perforin and granzymes to destroy bacterially-infected cells in HIV/SIV-infected individuals [9,43].

## 5. MAIT Cell Function during *Mycobacterium tuberculosis* Infection in Humans

Human MAIT cells have been examined for their ability to respond to mycobacteria, such as Mtb and BCG. BCG is commonly administered intradermally to children in TB-endemic areas as a vaccine, but its efficacy as a TB vaccine is variable and incomplete. Studies examining how human MAIT cells respond to mycobacterial stimuli is often limited to the peripheral blood, or ex vivo and in vitro experiments. As a result, data about MAIT cell frequency and function at the site of bacterial replication is often lacking, leaving holes in the picture of MAIT cell function in humans vaccinated with BCG or infected with Mtb.

Functional MAIT cells can recognize bacterially-infected cells and contribute to the resulting immune response. In patients with active TB disease, MAIT cell frequencies decline in the peripheral blood and are believed to migrate to the lungs [47]. However, while MAIT cell frequencies have been found to slightly increase in the lungs during TB infection [47], their measured response to antigens can vary based on the extent of TB disease and their location. For example, MAIT cells isolated from TB pleural effusions, which occur during severe active TB disease, had increased IL-2, IL-12 and IL-18-mediated IFNɣ, IL-17F, and granzyme B responses when compared to MAIT cells isolated from the peripheral blood after stimulation with Mtb lysates [48]. Additionally, MAIT cells isolated from the peripheral blood of active TB patients and stimulated in vitro with *Escherichia coli* had significantly decreased IFNɣ production when compared to healthy controls [49]. This seemingly large difference in MAIT cell function from patients with active TB may be a result of different sample types and stimuli used. These results also suggest that MAIT cells present at sites of severe TB disease may have a distinct functional phenotype when compared to those in the circulation, and that MAIT cell responses are likely specific to their environment and the antigens and cytokines they encounter.

BCG is a commonly administered vaccine in TB-endemic areas. However, it is not fully understood the mechanism by which MAIT cells respond to BCG stimulation. Peripheral MAIT cells from active TB patients stimulated with BCG ex vivo produced significantly more IFNɣ and TNFɑ when compared to healthy controls [20]. Interestingly, this observation was unique to BCG stimulation, as nonspecific PMA/ionomycin or *E. coli* stimulation did not lead to increased MAIT cell production of these cytokines in the active TB patients when compared to healthy controls [20]. However, whether the IFNɣ production of BCG-stimulated MAIT cells is dependent on direct detection of metabolites presented by MR1 is debatable. In contrast to Jiang et al., Suliman et al. reported that IL-12 and IL-18 were required for optimal IFNɣ production by BCG-stimulated MAIT cells, as antibody-mediated blocking of IL-12 and IL-18 completely eliminated IFNɣ production, whereas anti-MR1 blocking only partially reduced IFNɣ production [5]. Therefore, while these results support the hypothesis that human MAIT cells can recognize and respond directly to MR1 presented antigens, the MR1-independent mechanisms also contribute to the response of MAIT cells to mycobacteria.

There is a fair amount of evidence describing peripheral MAIT cell function during active TB infection, but a similar understanding of MAIT cell function in individuals with latent tuberculosis (LTBI) is less well defined [17,30,50]. Malka-Ruimy et al. (2019) reported a decrease in peripheral MAIT cell frequency in children with active TB when compared to TB-naive children or those with LTBI, and Paquin-Proulx et al. (2018) reported similar results in adult patients. In contrast, Suliman et al. (2020) found no differences in peripheral MAIT cell frequency between those with active and latent TB. Therefore, while MAIT cells in patients with LTBI typically resemble the MAIT cells of healthy individuals, this does not rule out the possibility that these cells may share phenotypes with MAIT cells from active TB patients. As we also do not fully understand how the immune response maintains latent TB infection and why some individuals never progress to active TB infection, it may not be an entirely fair comparison between active TB patients, healthy patients, and those with latent TB. Unfortunately, studies conducted in humans rarely include tissue-based analyses, as lung tissue and granulomas are difficult to assess from patients harboring latent mycobacteria. In order to more comprehensively study MAIT cells in tissues, animal models, such as NHPs, must be used.

## 6. MAIT Cell Function during *Mycobacterium tuberculosis* Infection in NHPs

The role of MAIT cells in mycobacterial infections, particularly within tissues, is not well understood in vivo. Assessing how MAIT cells respond directly to Mtb in humans can be difficult because the primary sites of Mtb-affected tissues, the lungs and granulomas, are often not accessible. Furthermore, there are often unknowns in human studies of Mtb infection, such as the length of time since the individual was exposed to Mtb and the concentration of bacilli present in that exposure. In contrast, the dose, route, and timing of infection of NHPs can be controlled, and researchers can euthanize animals at different time points to study disease and immune pathology in less accessible tissues, such as the lung, lymph nodes, and granulomas [26,51,52,53].

NHPs with TB develop a wide pathological spectrum similar to humans, ranging from LTBI to severe active TB [54]. While Mauritian cynomolgus macaques (MCM) show similar rates of TB disease progression to rhesus macaques (RM), Chinese cynomolgus macaques (CCM) display a lower rate of severe disease progression [11]. The decision of which NHP model to use for TB studies should be considered carefully, as each species has advantages and disadvantages for testing specific hypotheses about TB immunity and pathology.

Selection of the dose, route, and challenge strain of Mtb has been shown to impact the MAIT cell response detected and, therefore, should be chosen carefully. For example, in one study by Kauffman et al., MAIT cells from the bronchoalveolar lavage (BAL), granulomas, and lymphoid tissues of RM infected with the high-virulence Mtb Erdman strain did not display significant proliferation or activation as indicated by a failure to upregulate of ki67 or granzyme B, respectively, approximately 6 weeks following infection [55] In contrast, RM infected with the low-virulence Mtb CD1551 strain who developed active TB as of 9 weeks post-infection had a higher percentage of granzyme B positive MAIT cells in the BAL and PBMC when compared to animals with LTBI 11 weeks post infection, but diseased tissues were not analyzed [56]. In addition to the choice of bacterial strain, Buscan et al. [56] classified MAIT cells as TCRva7.2+CD161+, whereas Kauffman et al. identified MAIT cells with RM MR1 tetramers. Cumulatively, these studies emphasize how structural changes to a study, such as the mycobacterial strain used or the precise method for detecting MAIT cells, can produce wildly conflicting results about MAIT cells that are difficult to reconcile.

NHPs infected with Mtb are an ideal model in which to study MAIT cells in tissues with TB disease. At necropsy, individual granuloma lesions can be isolated, homogenized, and subjected to flow cytometry. In both Mtb-infected RM and MCMs, MAIT cells isolated from granulomas did not express high levels of activation markers, such as ki67, granzyme B, PD1, and TIGIT, and the frequency of MAIT cells present was not correlated to the bacterial burden [26,55]. Both of these studies euthanized animals at ca. 6–7 weeks after infection to maximize the chances of detecting activated MAIT cells. It is possible that characterization of MAIT cells at a later time point would have identified a stronger association between MAIT cell frequency and function and bacterial burden. MAIT cells from lesions at time points prior to 6 weeks would be incredibly difficult to isolate because the granulomas are very small. Therefore, while MAIT cells may be present in Mtb-affected tissues approximately 6 weeks after infection, these results do not provide evidence that they are effective mediators of Mtb control within granulomas just prior to development of the adaptive immune response.

MAIT cells detect mycobacterial antigens in the context of BCG vaccination, as indicated by increased levels of ki67 in both the blood and at the sites of intradermal BCG vaccination in RM [14]. However, as Greene et al. did not examine functional MAIT cell responses following BCG vaccination in this study nor did they examine MAIT cells in the lung tissues, it is hard to predict whether mucosal MAIT cells present in BCG-vaccinated individuals can offer protection against pathogenic Mtb. While there was a transient expansion in peripheral MAIT cells from Mtb-naive RM 2–4 weeks after IV-BCG vaccination, MAIT cell levels returned to baseline by 8 weeks post-vaccination [57]. Unfortunately, studies to determine whether MAIT cells are important for control of Mtb infection, such as MAIT cell depletion studies, are often not feasible in the NHP model given the current technology.

These limited studies of MAIT cells in Mtb-infected NHPs reveal how little we know about the role of MAIT cells in the tissues, particularly given the large spectrum of TB disease severity. Unfortunately, NHP studies are expensive and the cells available from individual granuloma lesions are limited. While these studies imply that MAIT cells are present in TB-affected tissues, there is no singular piece of data demonstrating that MAIT cells have an exceptional role in early control of Mtb replication. This does not mean that MAIT cells are agnostic to infection with Mtb, but it also does not mean that there is a singularly important role for MAIT cells in TB control. Thus, additional studies of MAIT cells in Mtb-infected NHPs that evaluate vaccines and therapeutics may help clarify their role.

## 7. Characterizing MAIT Cell Function in SIV-Infected NHPs Can Provide Insight into Our Understanding of MAIT Cell Function in HIV-Infected Humans

The features of SIV infection of NHPs closely recapitulate those observed in HIV infection of humans [58]. Because human acquisition of HIV is typically through mucosal routes, macaque studies involving intrarectal or intravaginal infections are of particular interest. SIV+ macaques are an excellent model system for assessing how an immunodeficiency virus infection impacts the function of MAIT cells in the periphery and at mucosal sites, given the similarities between human and macaque MAIT cells.

Viral infection can activate MAIT cells through cytokine-mediated mechanisms that do not rely on an interaction between the TCR and MR1 [6,8]. These include viruses such as influenza [59,60], HIV [8,61], dengue [60], and SARS-CoV-2 [62,63]. TCR-independent activation is mediated by IFNɑ, IL-12, and IL-18 produced by antigen-presenting cells and results in MAIT cell production of proinflammatory cytokines and the upregulation of activation markers [8,9,42,43,60,63]. Specifically, viral infection is associated with the upregulation of CD38, HLA-DR, PD1, and CCR5, and the downregulation of CD161 [24,25] on the surface of MAIT cells, as well as the production of IFNɣ, TNFɑ, and IL-17, all of which help combat infection [8,61].

The impact of HIV infection on MAIT cell frequency and phenotype has been difficult to dissect because the acute and chronic phases of disease impart different effects on MAIT cells. Early cross-sectional studies found a lower peripheral MAIT cell frequency in HIV+ individuals during chronic infection when compared to healthy, HIV-naive controls [22,61]. This observation was recapitulated in one cross-sectional NHP study when using the human MR1 tetramer, which is less effective at detecting MAIT cells than the newer macaque-specific MR1 tetramer [36]. While cross-sectional studies such as Tang et al. 2020 [43] and Leeansyah et al. 2013 [25] show fewer activated, cytokine-producing MAIT cells in the periphery during chronic HIV infection when compared to HIV-uninfected patients, longitudinal studies in humans [8,22] and NHP [26,27] found no decrease in peripheral MAIT cell frequencies when followed for up to one year post-infection. A contributing factor to these differences may be the duration of the study, as longitudinal and cross-sectional studies may be examining different disease states. While cross-sectional studies can provide a snapshot of the frequency and phenotype of MAIT cells, the time-consuming longitudinal studies are key to unraveling differences in the MAIT cell phenotype during both acute and chronic infection in a single individual. SIV disease progression in NHPs is accelerated relative to HIV disease progression in humans [64], making them further valuable for understanding the impact of immunodeficiency virus infection on MAIT cells.

From these longitudinal studies, it appears that SIV/HIV infection does not rapidly deplete peripheral MAIT cell frequencies, as earlier proposed. Ellis et al. (2020) and Lal et al. (2020) both reported activated MAIT cells in the periphery during peak viremia with decreased functional capacity during chronic HIV/SIV infection; however, Lal et al. does not make the distinction between MAIT cell memory subsets and bulk MAIT cells, whereas Ellis et al. further stratifies MAIT cells by central and effector memory phenotypes. Further, both studies report increased markers of proliferation, such as ki67, at peak viremia. Despite these similarities in frequency and expression of proliferative markers during acute viral infection, the effects of chronic viral infections on MAIT cells are less well understood. Subsequent studies, particularly those examining MAIT cell frequency and function in HIV- and SIV-affected tissues, will need to be performed in order to better understand how MAIT cells contribute to the antiviral response in chronic infection. 

## 8. Modeling HIV/Mtb Co-Infection in Macaques: Elucidating the Role of MAITs at the Sites of Mtb Infection

HIV co-infection can have a significant negative impact on the outcome of TB disease. TB is the leading cause of death in HIV-infected individuals, including HIV+ individuals on antiretroviral therapy (ART) [65,66,67]. We and others have observed that SIV infection of NHPs exacerbates TB disease burden and leads to increased tissue pathology and bacterial dissemination, particularly to extrathoracic sites and lymph nodes, regardless of whether SIV infection was before or after Mtb infection [52,68,69]. It has also been shown that SIV infection of NHPs with latent TB can reactivate the latent mycobacteria, leading to reactivated TB disease. These reactivators exhibit increased immune activation and proinflammatory cytokine production when compared to SIV+ macaques that did not experience latent TB reactivation [70].

Through the use of NHP co-infected with SIV and Mtb, we and others have examined the frequency and function of MAIT cells at tissue sites that are difficult to access in human patients. These sites include lung tissue, lymph nodes, granulomas, and BAL. In NHP studies, we can control for the order and timing of infection in macaques (SIV followed by Mtb, or Mtb followed by SIV) in order to determine how each disease impacts the outcome of the other. Thus, NHPs are an excellent model in which to understand how an immunodeficiency virus infection dysregulates the MAIT cell response to Mtb.

An early, provocative hypothesis was that SIV/HIV dysregulates MAIT cells, rendering them ineffective against Mtb. To test this hypothesis, MAIT cell frequency and function were measured from both SIV+ and SIV− naive macaques who were subsequently infected with Mtb. MAIT cell frequencies in the blood did not increase following Mtb infection in this model, and produced less TNFɑ in response to both *E. coli* and *Mycobacterium smegmatis* than SIV-naïve MCM [26]. However, in a different model of co-infection when RMs were first infected with Mtb and then co-infected with SIV, SIV infection led to a transient, but not statistically significant, increase in MAIT cells in the periphery [56], consistent with data from NHPs infected with SIV alone [26,27]. The MAIT cells present in the blood of these SIV-infected RMs expressed higher levels of granzyme B, with the highest levels of granzyme B in animals with more severe TB disease [56].

MAIT cells in Mtb granulomas of both SIV+ and SIV− naive NHPs are also far less frequent than other immune cell types, suggesting that MAIT cells do not overwhelmingly traffic to granulomas during Mtb infection [26,55]. Additionally, the MAIT cells present in the Mtb granulomas of SIV+ MCM were less likely to produce TNFɑ directly ex vivo, but displayed no differences in the expression of activation markers PD1 and TIGIT, the proliferation marker ki67, or the proinflammatory cytokine IFNɣ, when compared to granulomas isolated from SIV-naive MCM [26]. Within the lymph nodes of SIV/Mtb co-infected MCM, the MAIT cell frequency was lower in the lymph nodes relative to the lungs, as well as an impaired ability to produce TNFɑ when compared to the SIV-naive cohort [26]. Mtb-infected RM that reactivated TB disease following SIV co-infection did not have a significant increase in MAIT cell frequency in the BAL when compared to SIV-naive RM, but did have a higher amount of polyfunctional MAIT cells in the BAL when compared to animals that did not reactivate following SIV co-infection [56]. However, it is difficult to determine if MAIT cells are truly helping control Mtb infection, or if mycobacterial infection is leading to increased MAIT cell function without contributing to control of infection. While these studies utilized different species and strains of Mtb, they demonstrate that MAIT cells in co-infection have reduced TNFɑ production, as well as reduced frequencies of polyfunctional MAIT cells, at sites of active Mtb infection [26,56]. Cumulatively, these results suggest that despite MAIT cell function during TB infection, MAIT cells may not be helping to contain Mtb replication in the lungs and that SIV co-infection further disrupts MAIT cell function.

## 9. Gaps in Knowledge and Future Directions

While MAIT cells have garnered more attention in the last 10 years, there still remain outstanding questions about their role in host immunity to TB. Early reports from cross-sectional studies suggested that SIV/HIV infection led to a decline in MAIT cells [22,36,61]. These early data were contradicted by recent reports stating that there was an increase in MAIT cells during acute SIV/HIV infection [8,26,27]. Such disagreements in both human and NHP studies may be attributed to a variety of differences in experimental design. When focusing on NHPs, species-specific differences, particularly with RM, CCM, and MCM, may affect MAIT cell populations that are attributed to the variable severity of TB disease, as described in Maiello et al. (2018) [11], but this remains to be further investigated. Additionally, the utilization of different Mtb strains, such as the highly pathogenic Mtb Erdman versus the mildly pathogenic Mtb CD1551, and duration of study (6 weeks versus 12), may affect results, as exemplified by the Buscan et al. (2019) [56] and Kauffmann et al. (2018) studies [55]. As the body of literature grows, we will better be able to understand the nuances of the MAIT cell response to Mtb and SIV/HIV infection.

There also remain differences in how MAIT cells are phenotypically characterized. While the development of the MR1 tetramer has been helpful, this reagent is limited to the detection of MAIT cells specific for MR1 presenting 5-OP-RU. Other ligands can be presented by MR1 and the widely available MR1 tetramer presenting 5-OP-RU may not detect those MAIT cells specific for ligands like FO (7,8-didemethyl-8-hydroxy-5-deazariboflavin), PLI (photolumazine I), and PLIII (photolumazine III) [71]. In addition to using the MR1 tetramer, some people have defined MAIT cells by expression of CD161hi and TCR Vɑ7.2. This method excludes MAIT cells that downregulate CD161 following stimulation [24,25]. Additionally, while the MR1 tetramer can commonly detect MAIT cells with more diverse or uncommon TCRɑ chains, such as TRAJ12 or TRAJ20 in conjunction with the traditional TRAV1-2, rather than TRAV1-2/TRAJ33 [3,7], further development of reagents to more completely detect the broad TCR heterogeneity displayed by MAIT cells may increase our understanding of the function of this cell subset. 

Ultimately, MAIT cells provide a unique subset of innate-like T cells capable of recognizing and eliminating bacterial pathogens in the periphery and at mucosal sites. Since they are able to respond by direct TCR recognition of MR1 and through TCR-independent mechanisms through IL-12 and IL-18 signaling, they can be activated by microbes that do not synthesize riboflavin, such as viruses. Even though SIV/HIV infection can minimally dysregulate MAIT cells, there is limited data suggesting that this disruption leads to worsened TB disease. While we are not ready to eliminate a possible role for MAIT cells in Mtb-immunity, continuing to evaluate MAIT cells in NHPs who are co-infected with SIV and Mtb under different circumstances will be key to defining whether they should remain a focus of anti-Mtb immunotherapies.

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
