# Peer review of "Monkeying around with MAIT Cells: Studying the Role of MAIT Cells in SIV and Mtb Co-Infection"

_viruses, 2021, doi:10.3390/v13050863_

Round 1
Reviewer 1 Report
Overall, Moriarty and colleagues have produced a nicely constructed review on the literature surrounding MAIT cells, HIV/SIV infection, and Mtb (co)-infection. The review has a logical flow, is clearly written, and is a comprehensive overview of the field. In some places, for example line 237-246 and the 3rd paragraph of section 9, the authors have done a particularly good job of providing their expert opinion and summarizing a seemingly muddled literature. While at times the MAIT cell field feels quite flooded with review articles, this review is a very nice look at a focused, and under-summarized, aspect of MAIT cell biology.
Feedback:
- If possible, I would encourage the authors to use lines 237-246 as an inspiration for how other aspects of the MAIT cell/NHP literature can be nicely summarized. To someone outside the NHP field, many of the nuances of the various systems can make it very difficult to integrate the literature.
- A table of all of the NHP studies on Mtb, SIV, and Mtb/SIV could be a quite helpful resource to the reader. Including information such as NHP species used, infecting bacterial/viral strain, sampling schedule, etc. will really help non-expert readers.
- While I agree with the authors’ general claim that NHP models offer many unique benefits over both human and mouse studies, this repeated assertion becomes quite repetitive by the later portions of the review. Perhaps more “showing” the strengths of the NHP model by referencing specific studies will be of more benefit than “telling” of the benefits. For example, contrast lines 298-307 with 308-314.
- The title, while catchy, doesn’t really accurately reflect the literature. From the authors own conclusions, it is still unclear what role exactly MAIT cells might play in the control of Mtb infection. The title feels misleading in this regard.
- Line 145-147 is probably an over-statement. The authors acknowledge “the seemingly large difference in MAIT cell from patients with active TB is likely a result of different sample types and stimuli used”. It is not clear if MAIT cells actually have distinct functional phenotype based on the non-comparable stimulation conditions.
- Line 265-281 makes quite a big deal that longitudinal studies of acute infection do not recapitulate earlier findings that chronic HIV infection is associated with impaired MAIT cell frequency. Might the discordance simply reflect a difference in the stages of disease assessed between the longitudinal and cross-sectional cohorts? For example, Leeansyah et al., 2013 (ref 25) reported an average duration of infection as 85 months, while that groups more recent longitudinal study (Lal et al., 2020; ref 8) had only ~36 months of follow-up.
Concerns of factual accuracy:
- Throughout the manuscript, it is slightly confusing what aspects of MAIT cell biology being described have been discovered in NHPs versus humans. Care should be taken to better delineate what are unique phenotypes of NHP MAIT cells. For example, while the Mauritian macaques have a large MAIT cell phenotype with a Tcm phenotype (line 88-91), this is not a phenotype shared with human MAIT cells (e.g. Dias et al., 2017).
- Lines 261-264 state that MAIT cells produce IL-17 in response to HIV infection, but the references cited do not show HIV-induced production of IL-17 by MAIT cells.
- According to Reantragoon et al, 2015, the “standard” 5-OP-RU-loaded tetramers can detect the less common TRAV1-2/TRAJ20 and TRAV1-2/TRAV12 rearrangements with similar efficiency to Va7.2-TCR antibodies. So the data suggest that text on line 372-375 is an overstatement. While there are some rare MR1-restricted T cells that may not bind 5-OP-RU, the vast majority of MAIT cells can be robustly detected with the “standard” tetramer reagent.
Author Response
Response to reviewers
We thank the reviewers for their thoughtful and detailed reviews of this manuscript. We have adjusted the manuscript accordingly and have provided responses to each point, as highlighted below:
Reviewer 1:
Overall, Moriarty and colleagues have produced a nicely constructed review on the literature surrounding MAIT cells, HIV/SIV infection, and Mtb (co)-infection. The review has a logical flow, is clearly written, and is a comprehensive overview of the field. In some places, for example line 237-246 and the 3rd paragraph of section 9, the authors have done a particularly good job of providing their expert opinion and summarizing a seemingly muddled literature. While at times the MAIT cell field feels quite flooded with review articles, this review is a very nice look at a focused, and under-summarized, aspect of MAIT cell biology.
Feedback:
- If possible, I would encourage the authors to use lines 237-246 as an inspiration for how other aspects of the MAIT cell/NHP literature can be nicely summarized. To someone outside the NHP field, many of the nuances of the various systems can make it very difficult to integrate the literature.
We thank the reviewer for this comment.
- A table of all of the NHP studies on Mtb, SIV, and Mtb/SIV could be a quite helpful resource to the reader. Including information such as NHP species used, infecting bacterial/viral strain, sampling schedule, etc. will really help non-expert readers.
We thank the reviewer for this suggestion and have included this table.
- While I agree with the authors’ general claim that NHP models offer many unique benefits over both human and mouse studies, this repeated assertion becomes quite repetitive by the later portions of the review. Perhaps more “showing” the strengths of the NHP model by referencing specific studies will be of more benefit than “telling” of the benefits. For example, contrast lines 298-307 with 308-314.
- The title, while catchy, doesn’t really accurately reflect the literature. From the authors own conclusions, it is still unclear what role exactly MAIT cells might play in the control of Mtb infection. The title feels misleading in this regard.
We have adjusted the title accordingly to better reflect the literature.
- Line 145-147 is probably an over-statement. The authors acknowledge “the seemingly large difference in MAIT cell from patients with active TB is likely a result of different sample types and stimuli used”. It is not clear if MAIT cells actually have distinct functional phenotype based on the non-comparable stimulation conditions.
We have rewritten this statement to reflect the uncertainty between MAIT cell function and phenotype in these situations.
- Line 265-281 makes quite a big deal that longitudinal studies of acute infection do not recapitulate earlier findings that chronic HIV infection is associated with impaired MAIT cell frequency. Might the discordance simply reflect a difference in the stages of disease assessed between the longitudinal and cross-sectional cohorts? For example, Leeansyah et al., 2013 (ref 25) reported an average duration of infection as 85 months, while that groups more recent longitudinal study (Lal et al., 2020; ref 8) had only ~36 months of follow-up.
We have included a statement addressing the potential that longitudinal and cross-sectional studies may be examining different disease states, which is an important contributing factor.
Concerns of factual accuracy:
- Throughout the manuscript, it is slightly confusing what aspects of MAIT cell biology being described have been discovered in NHPs versus humans. Care should be taken to better delineate what are unique phenotypes of NHP MAIT cells. For example, while the Mauritian macaques have a large MAIT cell phenotype with a Tcm phenotype (line 88-91), this is not a phenotype shared with human MAIT cells (e.g. Dias et al., 2017).
This sentence has been edited to better reflect the distinction between Tem and Tcm phenotypes in humans and macaques.
- Lines 261-264 state that MAIT cells produce IL-17 in response to HIV infection, but the references cited do not show HIV-induced production of IL-17 by MAIT cells.
We have adjusted the text to indicate that IL-17 production is associated with viral infection, rather than a direct cause and effect relationship.
- According to Reantragoon et al, 2015, the “standard” 5-OP-RU-loaded tetramers can detect the less common TRAV1-2/TRAJ20 and TRAV1-2/TRAV12 rearrangements with similar efficiency to Va7.2-TCR antibodies. So the data suggest that text on line 372-375 is an overstatement. While there are some rare MR1-restricted T cells that may not bind 5-OP-RU, the vast majority of MAIT cells can be robustly detected with the “standard” tetramer reagent.
We have adjusted this sentence to better reflect the literature.
Reviewer 2 Report
I have gone through the whole manuscript and thought this is an excellent review article. The literature presented looks very convincing and updated. This review article discusses about the current knowledge surrounding MAIT cells in SIV and Mtb infection. Authors also made a point to describe the SIV infection impairs MAIT cell function during Mtb co-infection. I believe that this review article is interesting, and topic is beneficial to the scientific community.
Author Response
We thank the reviewer for their time and comments on our manuscript.